# Effects of experimental situation on group cooperation and individual performance: Comparing laboratory and online experiments

**Hiroki Ozono** [1] *, **Daisuke Nakama** [2]

**1** Faculty of Law, Economics and Humanities, Kagoshima University, Kagoshima, Japan, **2** Institute for Organizational Behavior Research, Recruit Management Solutions, Co. Ltd., Shinagawa-ku, Tokyo, Japan

* Hiroki.ozono@gmail.com

**Data Availability Statement:** All relevant data are within the paper and its Supporting information files.

**Funding:** HO was supported by the JSPS KAKENHI (18K13272). http://www.jsps.go.jp/english/. This

## Abstract

With the spread of online behavioral experiments, estimating the effects of experimental situations and sample heterogeneity is increasing in discussions of the generalizability of data. In this study, we examined how the experimental situations (laboratory/online) affected group cooperation and individual performances. The participants were Japanese university students, randomly assigned to laboratory or online experiments. For the group cooperation task, they were asked to perform the public goods game with or without punishment, but no effect of the experimental situation was found both for cooperative and punitive behaviors. For the individual tasks, participants were asked to perform tasks including a creative task and a dull task. We manipulated the presence or absence of an external incentive. As a result, there was no significant difference between the experimental situations with one exception: only in the laboratory situation was the performance of the difficult creative task lower in the presence of an external incentive. Furthermore, we conducted as an additional experiment using the same treatments for a Japanese online-worker sample. This sample was less cooperative in the public goods game than the student sample, both with and without punishment. In addition, the presence of external incentives facilitated performance of the online-worker sample only for the dull task. We discuss the similarities and differences with previous studies that examined the effects of experimental situations and sample heterogeneity, and the implications for remote work in the real world.

## Introduction

In typical behavioral experiments in economics and psychology, university students participate in and perform a task in a laboratory, mainly because of the ease of providing a controlled environment. Although behavioral experiments are a powerful way to control factors and identify causality, there is repeated criticism of generalizability concerning whether the same trends can be obtained outside the laboratory and in people other than university students [1,

funder had no role in study design, data collection and analysis, decision to publish, or preparation of the manuscript.

**Competing interests:** The authors have declared that no competing interests exist.

2]. Due to the recent development of the Internet and devices that allow a heterogenous sample, i.e. a more general sample, to participate in online behavioral experiments from outside a laboratory, increasing numbers of studies have attempted to address this criticism. Such studies consider the generality of behaviors and the reliability of data by comparing laboratory experiments using university students and online experiments using a heterogenous sample.

It should be noted that comparing university students in a laboratory with a heterogenous sample online makes it difficult to separate the effects of the sample heterogeneity (i.e., university students/heterogenous sample) from the effects of the experimental situation (i.e., laboratory/online). In this study, we focus primarily on the effects of the experimental situation for two reasons. The first is that there are relatively few studies on the effects of the experimental situation with appropriate controls [3–7], although there are many studies on the effects of the sample heterogeneity [6, 8–18]. In regard to studies of sample heterogeneity effects, researchers have shown similar qualitative patterns and correlations in both samples, although there are quantitative differences such as less cooperativeness for university students [14–18]. However, studies on the experimental situation are rare and the results are mixed, as we review in the next section. Thus, we believe that more research on the experimental situation is needed. The second reason is a change in the work environment in the real world, that is, the increase in remote work. Specifically, the COVID-19 pandemic has accelerated the spread of remote work [19, 20], and this trend will continue in the future. In this case, the same employee works either in the office or at home. Thus, it can be said to be parallel to studies of the experimental situation: the same sample (in this case, university students) participated in either laboratory or online. Examining the differences in behaviors between laboratory and online will provide useful insights into the effectiveness of remote work and what kind of tasks and factors cause differences in performance. To obtain broader implications for work in the real world, we will focus both on the effects of group work (prosocial behavior) and individual task performance to examine the effects of the experimental situation.

Previous studies have proposed two theories concerning why the experimental situation affects behavior: the observer effect and social distance theory. The observer effect [6, 21] is caused by being observed by the experimenter in the laboratory, and this predicts that participants are more prone to socially desirable behavior and more sensitive to social pressure because they are intentionally or unintentionally concerned about the social evaluation by the experimenter. Social distance theory [4, 22, 23] predicts substantially different behavior, such as prosocial behavior, over a disembodied and distant network compared with close physical and emotional proximity. In the laboratory situation, participants can see each other before and after the experiment and their physical distance is closer during the experiment than it is online, which may facilitate prosocial behavior and impose more social pressure. Ultimately, the two theories lead to identical predictions and are conceptually similar. In this study, we do not focus specifically on the differences between these theories, but simply trace the discussions in previous studies that compared the laboratory with the online situation.

### Research question 1: How does the experimental situation affect group cooperation and punishment behavior?

Some experiments have examined the effects of experimental situations on prosocial behavior [3–7, 24, 25]. Among them, five studies used random assignment from the same sample group to eliminate selection and sample biases. The results are mixed. Two studies found no significant differences [3, 6] and one found, contrary to predictions, that participants online tend to be more prosocial [4], and two studies are consistent with the prediction that online participants tend to be less prosocial [5, 7]. These studies used games commonly used to measure

prosocial behavior, such as the public goods game (PGG), the trust game, and the dictator game (with one exception; the study by Schmelz and Ziegelmeyer [5] has the structure of a principal–agent problem, which captures a conflict in priorities between an agent and their principal [26]). Thus, the inconsistent results on the effects of the experimental situation warrant further research studies. We should note that these inconsistent results might be due to differences in sample populations, and it might not be appropriate to simply pool them together. However, all these previous experiments were conducted using university students in Western countries, and it is difficult to obtain other detailed information of the sample groups, so we cannot discuss this point further.

Moreover, we emphasize that no studies have examined punishment behaviors. Previous studies suggested that punishing free riders, who obtain the benefits of cooperation but do not cooperate themselves, is effective and the experiments have shown that punishment maintains group cooperation [27–29]. Arechar et al. conducted a laboratory experiment with university students and an online experiment with an online-worker sample using Amazon Mechanical Turk (MTurk) [8]. They found no qualitative differences in the effect of punishment on maintaining cooperation. However, as mentioned above, the factors of sample heterogeneity and experimental situation were confounded in their study and the influence of the environment on punitive behavior has yet to be examined independently.

The experimental situation might affect punishment behavior. Because punishment involves damage to others, the online participants with their anonymity would be less likely to hesitate in punishing than in face-to-face laboratory situations where physical distance is close. For this reason, punishment behavior may increase in online situations and result in achieving group cooperation. Previous studies comparing laboratory and online situations have dealt with prosocial behavior such as altruistic and trust behaviors but, to the best of our knowledge, none have directly investigated punishment behaviors. In this study, we examine whether the effects of punishment on group cooperation differ across experimental situations.

## Research question 2: How does the experimental situation interact with the effect of external incentives on individual performances?

It is important to know how performance of individual work is affected by the situation, especially when discussing effectiveness of remote work. Ariely et al. showed that external incentives have a positive effect on performance of dull tasks that do not require creativity, while external incentives have a negative effect on performance of tasks requiring creativity [30]. They argued that the latter is caused by "choking under pressure" due to excessive incentive. These effects might vary depending on the experimental situation (i.e. laboratory or online). This is because the laboratory situation is more likely to be tense because of the closer physical distance from the experimenter and other participants, and greater concerns about monitoring and evaluation, which leads to higher social pressure. In contrast, in the online situation, the participants can be more relaxed in doing their tasks. Thus, the "choking under pressure" effect might be stronger in the laboratory: the effect of external incentives on decreasing performance for creative tasks might be greater in the laboratory.

Dutcher examined how performance on dull and creative tasks changed depending on the experimental situation [31]: performance on dull tasks was better in the laboratory, and creative task performance was better online. These results can be considered to reinforce that the laboratory situation induces more pressure, leading to the opposite effect on the dull and creative tasks. However, this study was conducted only under conditions with external incentives. We examine how the presence or absence of external incentives affect performance in both laboratory and online. In addition, we examined the task which requires convergent thinking

(anagram task [32, 33]) as well as the creative task which requires divergent thinking (remote association task [34, 35]) and the dull task which does not require specific thinking (inverse-anagram task; as subsequently explained) to investigate the effect of situations and external incentives on various tasks. This investigation will be useful in determining the appropriate settings for experimental research. Furthermore, the findings will practically help office managers in the real world when considering working conditions for remote workers.

We also conducted an additional experiment with a Japanese online-worker sample via a crowdsourcing service, which is more heterogeneous than a student sample. This allowed the comparison between the behaviors of students online and a heterogenous sample and examination of the effects of sample heterogeneity on group cooperation and punishment as well as on individual performance. We emphasize the significance of comparing a heterogeneous sample with a student sample. Exadaktylos et al. [36] point out that while bias in demographic factors such as gender and age can be statistically controlled, student bias can occur due to various non-controllable variables. They collected data and found that although student bias did exist, it was only at 5% level for a single effect. It is important to accumulate data to examine the existence of such bias, and our study will help in this regard. We collected the online-worker sample from the Yahoo crowdsourcing service (hereafter, YCrowd), which is one of the largest crowdsourcing services in Japan, as a heterogeneous sample. This is because many research studies uses crowdsourcing services such as MTurk due to ease of data collection, and a comparison between student and online-worker samples provides beneficial insight for researchers. It should be noted that collecting data for an online-worker sample in a laboratory situation might be more integrative and comprehensive. However, it is difficult and costly to obtain these data in the COVID-19 pandemic situation and these data are elusive for the purposes of our study. Thus, we did not collect data of the online-worker sample in a laboratory situation.

## Main experiment

### Methods

We randomly assigned the participants from the university student pool to either laboratory or online experiments. For the group task, groups of three participants were formed and performed the PGG without punishment, followed by the PGG with punishment. It may be better to counterbalance the order of the PGG with and without punishment. But, following the study by Arecher et al. [8], we did not include the counterbalancing measures in this study. In addition, no evidence of order effects as found by Fehr & Gächter [37] supports this setting. After that, all participants performed three individual tasks. The contingent and fixed-pay conditions were randomly assigned among the participants to manipulate the presence or absence of incentives. This study was approved by the Ethics Review Committee of the Faculty of Economics, Law and Humanities, Kagoshima University, Japan.

**Participants.** We sent a recruitment email to undergraduate students at Kagoshima University who were registered in the psychology experiment participation pool. Only students who could register for both online and laboratory experiments were eligible to apply. We randomly assigned them to either situation. There were 13–15 participants per session, with 51 (24 women, 26 men and 1 unknown) in the laboratory experiment (mean age of 20.04, SD = 1.25 years) and 56 (36 women, 18 men and 2 unknown) in the online experiment (mean age of 20.32, SD = 1.30 years). We did not find any significant differences between online and laboratory experiments including age and gender (see S1 Table in S1 File). Each session took an average of approximately 80 minutes. Average earnings in this study sample were 1214 yen; currently, the minimum hourly wage in Kagoshima is approximately 800 yen. Thus, this

average reward amount of 1214 yen for a single 80-minute-session is close to the amount students commonly receive for their part-time work. To ensure the same procedure in the laboratory and online, the reward was paid by sending an Amazon gift card to each participant's email address. We conducted the experiment in November 2020. During this period, COVID-19 was prevalent, so we had to consider potential effects on those who came to the laboratory. Cancellations for the experiment were 4 out of 55 for the laboratory condition and 3 out of 59 for the online condition, and thus we concluded that there was no appreciable bias.

**Group tasks.** *Public goods game without punishment.* The PGG and peer punishment were designed based on Fehr and Gachter [27]. First, a group of three was formed, and the session was repeated for 10 periods with the same members. In the beginning of each session, 20 points were given to each participant as an endowment, and participants simultaneously decided how many of these points they would give to the group. The points they kept were added to the amount they earned. After all participants had made their decisions, the total contribution was multiplied by 1.5 and distributed equally to all three members regardless of their contribution. Each contribution and the amount to be distributed was immediately presented to the members.

*Public goods game with punishment.* The PGG with punishment was conducted after PGG without punishment, and also repeated for 10 periods with the same three members. The conditions were the same as for the PGG. After each contribution was presented, the participants had an opportunity to reduce the amount of the other members. For every one point used to reduce another member, this other member was reduced by two points. Each member could use up to a total of 10 points for reduction. All members decided simultaneously who to reduce and by how much; after the decision was made, they were shown how much they had spent on the reduction and how much they had been reduced in total.

**Individual tasks.** *Anagram task.* In this task, participants rearranged a randomly sorted string of five characters (Hiragana: Japanese phonogram) into an understandable word. Anagrams have been widely used as a task for measuring convergent thinking [32]. The Japanese version of the anagram task is available in a database provided by Ichimura et al. [33]. Seventy-two questions were randomly selected from the database for the present study (see the questions in the S1 Data).

*Remote association task.* A remote association task is a task that measures creative and divergent thinking [34, 35]. Participants are shown a set of three words and asked to provide a single word that is associated with the three-word set (e.g., "ice" from cream/skate/water). In our study, we used the database of the Japanese version [38] of the Compound Remote Association Task [39], which is widely used because the answers are uniformly determined. In the Japanese version, three Kanji characters (Japanese ideographs based on Chinese characters) are presented to the participants, and they are required to answer one Kanji character that is linked to all three. In this experiment, we randomly selected 40 questions from the database and used them.

*Inverse-anagram task.* We developed an inverse-anagram (hereafter, in-anagram) task to compare performance with the standard anagram task. In the in-anagram, participants were presented with a five-letter word and asked to randomly sort and input the word according to the five numbers following it. For example, when the word is presented as "THINK (34512)", the third letter, "I", should be put in the first position, the fourth letter, "N", should be put in the second position, and so on. Then, "INKTH" is the correct answer. Because Japanese participants were our targets, we used five Hiragana characters instead of the alphabet. This task, rearranging the letters according to the instructions, is quite simple and does not require any creative thinking. However, it is identical to the anagram task in the sense that the participants need to rearrange the letters and input characters. For the five-letter words used in the

experiment, 100 words were randomly selected from the database of Ichimura et al. [33] excluding the 72 questions used in the anagram task explained above.

**Questionnaires.**   After all tasks were completed, the participants answered the post-questionnaire. In addition to demographic items such as gender and age, the participants were asked to answer how much they enjoyed each task on a five-point Likert-scale in order to measure their intrinsic motivation concerning the individual tasks. They also completed measures for social value orientation (SVO) using sliders [40]. Other questions such as the state of tension during the experiment were asked; however, no particular trends worthy of report were detected (see S1 Table in S1 File). All items and data are available (see S1 Data).

**Procedure.**   *Setting of laboratory experiment.* The participants visited the laboratory at the designated time, and after taking their temperature and checking their physical condition to prevent infection, they entered the booths separated by partitions. Booths were more than one meter from each other and, although they could not see other participants' faces, they could see parts of others' bodies and hear others' typing sounds (see the video of the laboratory during the experiment, https://youtu.be/rM9uFSIjCC4). There was only one male experimenter, and he always sat at the desk in front of the laboratory during the experiment. After the participants sat in their respective booths, they were asked to fill out an electronic consent form. After all the consents were confirmed, the experimenter orally provided general instructions to participants that they should access the link for the experiment and follow the instructions, and that if they had any questions, they could ask the experimenter using the chat function of Zoom, a proprietary video teleconferencing software program, that was installed on each PC. Zoom was used for the questions to make the setting as similar as possible to the online situation. Questions from participants were rarely encountered in either the laboratory or online experiments, requiring only a few simple confirmations.

*Setting of online experiment.* Participants were asked to enter the Zoom virtual room at the designated time, which they were notified of via email. They were asked to enter via PC, not smartphones, to be consistent with the laboratory experiment. Once they entered the virtual room, they were asked to fill out an electronic consent form. After consent of all participants was confirmed, a verbal announcement about the experiment was made by the experimenter. The content of the announcement was the same as in the laboratory experiment.

*Flow of the experiment.* We used oTree [41] to control the experiment. After entering the experimental program via oTree, the process was the same for both laboratory and online experiments (see Supplementary method in S1 File for the detailed instructions for the participants).

First, an explanation of PGG was given, followed by a confirmation test to ensure that participants understood the payoff of the game. For all the explanations and confirmation tests, the participants read the text presented on the screen, and there was no verbal intervention by the experimenter. They were informed that the earning points in PGG would be converted to 1 yen per point. Then, the PGG without punishment was repeated 10 times by three group members which were randomly formed. Next, additional explanations for the PGG with punishment were given. After the confirmation test, the PGG with punishment was repeated 10 times with the same members.

To ensure smooth running of the experiment, a time limit was set for each decision. Contribution decisions needed to be made within 40 seconds, and if the time limit was exceeded, the full 20 points were automatically contributed. We set the full contribution as the default to ensure that the remaining participants would not be disadvantaged financially. Punishment decisions needed to be made within 80 seconds, and if the time limit was exceeded, automatically the participant used no amount for punishment. Groups that had more than one case of

exceeding the time limit were to be excluded from the analysis, but we found no such cases in the experiment.

Once the group task was completed, we moved to the next three individual tasks. In the individual assignments, there was no need to wait for the other participants; they proceeded at their own pace. The order of the three tasks was either anagram–in-anagram–remote association task or in-anagram–anagram–remote association task. Since anagram and in-anagram tasks were similar for the participants, we regarded them as one unit to simplify the instruction process and to enhance the smooth transition between tasks.

The flow of each individual task was the same. First, the tasks were explained to the participants, and then they were given two trial questions to answer with a time limit of 1 minute. If they got them wrong, they were told the correct answers and asked to answer the same questions again for confirmation. For the anagram and remote association tasks, there were no participants who made mistakes twice, so the data of all participants were used for the analysis. For the in-anagram task, five participants who gave wrong answers twice were excluded from the analysis. In the experimental sessions, half of the questions we selected were presented on the screen at the same time, and the participants were asked to answer as many questions as possible within 4 minutes. When 4 minutes elapsed, the other half of the questions were automatically presented with no break and the participants continued to answer them. The participants were told that they would receive 100 yen regardless of the number of correct answers in the fixed-pay condition. In the contingent-pay condition, they would receive 8 yen for each correct answer in the anagram, 3 yen for the in-anagram, and 15 yen for the remote association task. This rate was set so that the average amount would be about the same as the fixed-pay condition, based on the results of the preliminary experiment: 10 undergraduate students in a psychology course from the same university participated in the preliminary experiment and completed all tasks without monetary reward.

After these three tasks were completed, the participants answered a questionnaire that included SVO and demographic variables. The participants were allowed to leave the room after they had finished answering the questions. The final reward was determined by calculating the earned amount according to the experimental results and the presented rate, plus the basic participation fee of 500 yen.

## Results and discussion

**Group tasks.** We found no exceeding of the time limit in the experiment, so all data could be used in the analysis. However, three participants in the laboratory and five participants online could not form a group of three members because the number of participants gathered in the sessions was not a multiple of three. In these cases, the participant played PGG with an automatic program, that contributed the full 20 points in PGG and punished nothing. All 16 groups (48 members) in the laboratory and 17 groups (51 members) online were used in the analysis.

The PGG contribution and the total amount of punishment use in the laboratory and online experiment are shown in Fig 1. First, to examine whether there was a difference in the PGG contribution between the laboratory and online, we calculated the average contribution throughout 10 periods for each group and conducted a Mann–Whitney U-test. There was no significant difference between the online and laboratory conditions in the PGG either without ($U = 119$, $Z = 0.59$, $p = 0.552$) or with punishment ($U = 142$, $Z = 0.23$, $p = 0.815$). We also performed a Mann–Whitney U-test for each period, but none significantly differed (all p-values $> 0.100$). In addition, there was no significant difference in the total amount of punishment use between the laboratory and online ($U = 111$, $Z = 0.87$, $p = 0.387$). Thus, we found no

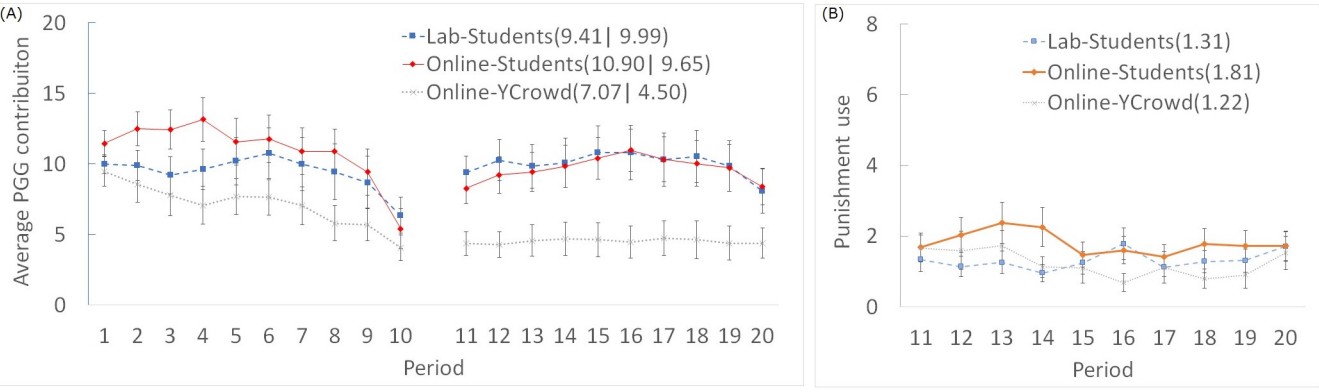

**Fig 1. (A) The PGG contribution and (B) use of punishment over time.** Numbers in parentheses are the mean contributions or punishment use in each experimental condition. Error bars indicate standard errors (clustered at the group level). "Online-YCrowd" indicates data of the Yahoo crowdsourcing sample collected in the additional experiment.

significant differences in the PGG contribution and punishment behavior between laboratory and online experiments. This is consistent with previous research showing that prosocial behavior is not attenuated in remote situations when participants are randomly assigned from the same sample group [3, 4, 6]. Furthermore, we found the same trend in repeated PGG with punishment. The multilevel regression analysis also showed the same tendency (see S2–S4 Tables in S1 File).

Fig 1 shows that the effect of punishment on increasing PGG cooperation appears to be weak. There was no difference in the average PGG contribution throughout 10 periods between with and without punishment conditions, either in the laboratory (signed-rank sum test: $T = 67$, $Z = 0.03$, $p = 0.979$) or online ($T = 53$, $Z = 1.09$, $p = 0.276$). However, there were some effects of punishment. First, there was a significant increase in contribution in the first period with punishment compared to the final (10th) period without punishment (Laboratory: $T = 21.5$, $Z = 1.92$, $p = 0.056$; Online: $T = 27.5$, $Z = 2.07$, $p = 0.039$). In addition, in the PGG without punishment, the contribution was lower in the last than in the first period, indicating a decay in cooperation (Laboratory: $T = 24$, $Z = 2.02$, $p = 0.044$; Online: $T = 8$, $Z = 3.08$, $p = 0.002$). However, in the PGG with punishment, there was no significant difference in contribution between the first and final periods, indicating that cooperation was maintained to some extent (Laboratory: $T = 32$, $Z = 1.26$, $p = 0.209$; Online: $T = 71$, $Z = 0.24$, $p = 0.813$). These results imply that the introduction of punishment had some effect in promoting and maintaining group cooperation in the online situation as well as the laboratory.

One reason for the weak effect of punishment in our experiment may be because the punishment efficiency was 2, which is less than the 3 in most previous studies [42]. As mentioned in the Methods section, each use of 1 point for punishment allows subtraction of 2 points from the other member, which amounts to a punishment efficiency of 2. Although we established this parameter to avoid a ceiling effect of contribution in PGG due to excessive power of the punishment, this low efficiency might weaken the punishment effect more than we expected. Another reason may be attributed to unique characteristics of Japanese culture; Funaki [43] conducted a PGG with punishment similar to that of Fehr and Gachter [27] for a Japanese university student sample and found that the effect of punishment was very weak, consistent with our results. Funaki points out that anti-social punishment, especially perverse punishment (i.e. punishment from the lowest cooperator to the highest cooperator), may inhibit cooperation, but the reason for this tendency in Japan was unclear and requires future research studies.

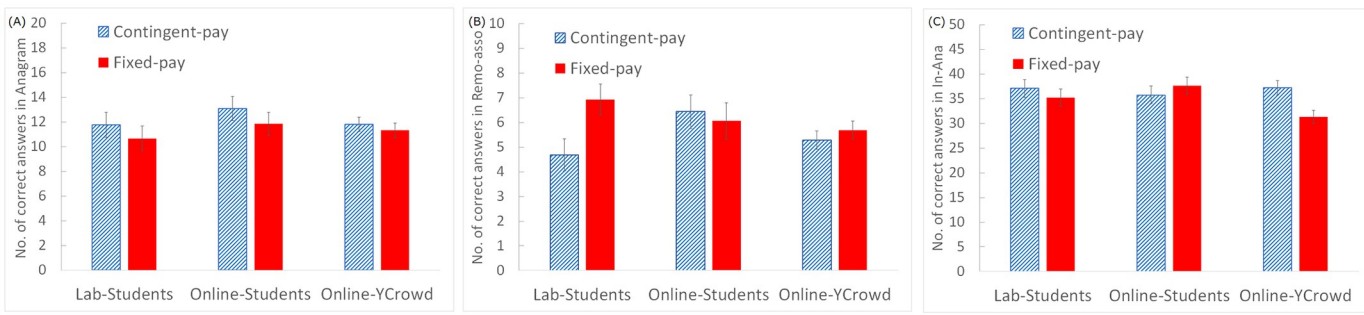

**Fig 2. Task performance (number of correct answers) for (A) anagram, (B) remote association, and (C) inverse-anagram tasks.** Error bars indicate standard errors. "Online-YCrowd" indicates data of the Yahoo crowdsourcing sample collected in the additional experiment.

**Individual tasks.** In one online session, the time for PGG was too long because of long consideration by a specific participant, so it was difficult to finish all the tasks within 90 minutes, which was the maximum time that we announced in advance. Thus, we decided to abandon the remote association task for this one session. For this reason, 14 data points are missing only for the remote association task online.

Fig 2 shows the results for the number of correct responses for the three tasks. We performed a between-subjects two-factor ANOVA for each task, situation (laboratory/online) × presence of incentive (contingent/fixed pay), with the number of correct responses as the dependent variable.

For the anagram task, all main effects and interactions were not significant (all F-values < 1.64, all p-values > 0.100). For the remote association task, the main effects were not significant (all F-values < 1.55, all p-values > 0.100), but the interaction was marginally significant (F(1,98) = 1.64, partial $\eta^2$ = 0.041, p = 0.056), and the simple main effect test showed that the number of correct responses decreased with contingent-pay only in the laboratory condition (t(89) = 2.47, d = 0.93, p = 0.015) (see Fig 2(A)). For the in-anagram task, all main effects and interactions were not significant (all F-values < 1.09, all p-values > 0.100). It should be noted that the Mann–Whitney U-test showed the same trend, that is, significance occurred only for the laboratory condition of the remote association task. In addition, the regression analysis controlling age, gender and the order of the tasks also showed the same trend (see S5–S7 Tables in S1 File).

The results obtained in the remote association task are consistent with the findings of Ariely et al. [30], which indicated that high incentive pressures reduce performance in creative tasks. This tendency was observed only in the laboratory situation, consistent with the study by Dutcher [31]. It is possible that the higher social pressure induced by the other participants and the experimenter in the laboratory, combined with the external incentive, caused the lower performance. In contrast, the anagram task, which needs convergent thinking showed no incentive effect. The anagram task requires insight but not creative divergent thinking, and only requires manipulation of the material (letters) presented. Therefore, it may be less susceptible to the influence of such social pressure. However, this difference might have arisen from the task difficulty. The rate of correct responses for the anagram task was 81%, while that for the remote association task was 35%, indicating that the remote association task was far more difficult. Therefore, social pressure may have strongly affected performance in the remote association task. This low performance of the remote association task in the laboratory might be attributed to the difficulty of the task rather than the creativity. The rate of correct responses for the in-anagram task was 89%; this was as easy as the anagram task, and we found no

incentive effect. Determining the cause of low performance in the remote association task will require further research. In addition, we note that it is difficult to conclude which factor is more influential—the presence of an experimenter or other participants. This awaits future research.

Another interesting point is that a positive effect of external incentive was not observed in the in-anagram task (a dull task). One possibility is that the in-anagram task was also sufficiently motivating internally. However, the question for measuring internal motivation, in which participants were asked to rate enjoyment of the task itself on a five-point scale in the post-questionnaire, indicated that the rating for the in-anagram task (M = 2.79) was significantly lower than for the anagram task (M = 3.15: t(92) = 3.02, d = 0.28, p = 0.010 after Bonferroni correction) and the remote association task (M = 3.11: t(92) = 2.21, d = 0.25, p = 0.089 after Bonferroni correction). Thus, the in-anagram task was less likely to arouse intrinsic motivation. Another possibility is that the participants might be highly motivated to perform the task seriously regardless of external and internal incentives. The participants in this experiment were university students who responded to the recruitment mail, adjusted their schedules, and then participated in the experiment. Thus, they might be highly motivated to do tasks. In addition, the experiment was conducted by a professor, who has the authority to evaluate students and is in a position of higher social status: older and more influential. Therefore, they might perform the tasks as seriously as possible, regardless of monetary incentives. If this were the case, we would expect that the effect of the presence of incentives would be more likely to occur in online-worker samples, who are free from the influence of authorities.

**A post hoc power analysis.** Finally, we conducted a series of post hoc power analyses. For the group task, the effect sizes for differences in the PGG contribution and the total amount of punishment were moderate to small (the PGG contribution without punishment, d = 0.26; with punishment, d = 0.05; and the amount of punishment, d = 0.38). For the evaluation of effect sizes, we follow Cohen's criteria [44]. We used G*power 3.1 to obtain post hoc power on the basis of the observed effect sizes and sample sizes [45]. The $\alpha$ was set to 0.05 and the estimated powers were the following: PGG contribution without punishment, 0.11; with punishment, 0.05; and the amount of punishment, 0.18. These powers were small mainly due to the small effect size. If we ensure 80% power with these effect sizes, we would need very large samples: 472 groups (1416 participants) are needed for the PGG contribution without punishment, and 234 groups (702 participants) for the punishment use. These results provide support that the differences between the laboratory and online situations do not require serious consideration. For the non-significant main effect and the interactions of the individual tasks, the effect size was low (partial $\eta^2$ values < 0.016). and the estimated power was also low (powers < 0.26). We note that the partial $\eta^2$ cannot be simply evaluated from the numerical values, but in the case of this study, the numerical values were at most 0.001 different from the $\eta^2$, so it was not a problem to evaluate from the numerical values of partial $\eta^2$. For the significant interaction in the remote association task, the effect size was 0.041, which is a moderate effect size, but the estimated power was 0.50, which is not sufficient. Since this study was conducted in the pandemic of COVID-19, the obtainable sample size was uncertain and prior sample size determination was difficult. Future studies should address this issue.

## Additional experiment

We collected data on the same tasks for an online-worker sample in Japan. The purpose of this additional experiment was to examine the effects of sample heterogeneity by comparing the data with our online experiment using the student sample. In the group cooperation task, we examined whether the degree of cooperation and the effects of punishment differed across the

sample; and in the individual performance task, we examined whether the effects of incentives, mainly in dull tasks, were more likely to occur in the online-worker sample. The participants in the main experiment participated in both the group cooperation and the individual tasks; however, in the additional experiment we separately conducted the group cooperation task and the individual tasks because it would be difficult to keep the online-worker sample for a long time.

## Methods

**Participants.**   We posted a notice of participation on YCrowd website, and registered users participated in the experiment. In Japan, MTurk is not popular, and one of the largest crowdsourcing services is YCrowd. There are about 700,000 registered users (as of January 2021) consisting of a native Japanese-speaking population. The YCrowd users who accessed via PCs, not via smartphones, could participate in these experiments.

For the group cooperation task, 16 out of 99 participants dropped out in the middle of a task, and groups with more than one dropout were excluded from the data analysis. As a result, 63 samples in 21 groups were used in the analysis. For the individual tasks, 31 out of 200 participants dropped out in the middle of a task, and 169 samples were included in the analysis. We conducted the experiment from December 2020 to January 2021. The total sample used in the analysis was 232 (56 women, 171 men and 5 unknown; mean age of 46.33, SD = 9.86 years). These demographic factors differed from the student samples (see S1 Table in S1 File), but after controlling for these factors, we found similar tendencies (see S2–S7 Tables and S1-S3 Figs in S1 File).

**Tasks.**   All tasks were exactly the same as those in the main experiment.

**Procedures.**   The flow of each task was the same as the main experiment, but there were three differences as follows. First, in the main experiment, the participants entered the Zoom screen, where they were given a verbal explanation, and then proceeded to the oTree experiment; however, in the additional experiment, the participants immediately proceeded to the oTree experiment for easy participation. Second, for individual tasks, half of the participants did the remote association task first and the other half did it last for counterbalance, although all participants in the main experiment did the task last. This was to examine whether there was any effect of the remote association task always being last in the main study, but we found no such order effect for the task (see S5–S7 Tables in S1 File). Third, we set the participation fee and rate to be similar to the market rate for other tasks on YCrowd. Specifically, the participation fee was set at 30 yen, and the PGG rate was set at 0.2 yen per point. For individual tasks, they could earn 35 yen regardless of the number of correct answers under the fixed-pay condition, and under the proportional-pay condition, they could earn 3 yen for anagram, 1 yen for in-anagram, and 5 yen for remote association per correct answer. We calculated the rewards after the experiment and issued each of them digital currency points that were equivalent in value to cash payments.

## Results and discussion

**Group tasks.**   We analyzed the differences in average contribution throughout the 10 periods in PGG between the YCrowd participants and the university students in the online experiment. Mann–Whitney U-test showed that the YCrowd participants contributed less than the university student participants in the PGG without punishment (U = 137.5, Z = 2.26, p = 0.024) and with punishment (U = 111.5, Z = 2.89, p = 0.004). There was no significant difference in the total amount of punishment use between the student and YCrowd participants

(U = 175, Z = 1.35, p = 0.177). The multilevel regression analysis after controlling for demographic factors showed a similar tendency (see S2–S4 Tables and S1 and S2 Figs in S1 File).

In addition, for the YCrowd participants, we found that the contribution was lower for with punishment than without punishment condition (signed-rank sum test: T = 26, Z = 3.09, p = 0.002). Furthermore, there was no significant increase in contribution in the first period with punishment over the last period without punishment (T = 72, Z = −0.19, p = 0.850). However, punishment had some effect on the decay in cooperation for YCrowd participants: the PGG contribution decreased from the first to the last period in the PGG without punishment (T = 12, Z = 3.45, p = 0.001) but not with punishment (T = 58, Z = −0.49, p = 0.623).

Thus, results of the additional experiment showed that YCrowd participants were less cooperative and that there was a very weak effect of punishment. This means that the YCrowd participants in our additional experiment were less motivated to achieve cooperation than university student participants. This difference was due to the differences in samples. Many previous studies have reported that heterogeneous samples are more cooperative [8, 14–18], so the opposite result in our study is interesting. The reason for this may be a cultural characteristic of Japan, but a previous study conducted in Japan also showed that the heterogeneous sample was more cooperative than university student samples [16]. This raises the possibility that our results are not consistent even within the confines of Japan, thereby leaving our results open to various interpretations. This difference might be due to the difference in monetary incentives rather than heterogeneity of the sample—the lower incentive in the YCrowd experiment might lead to a lower cooperation level. Some previous studies suggest that payment scale is not very important [46, 47], but some studies suggest that the payment scale affects the results quantitively, but not qualitatively [48, 49]. Although we were unable to find a study that suggests a lower incentive leads to a lower cooperation level, further investigation is needed to clarify this point.

**Individual tasks.** Twenty-one participants for the in-anagram task and three participants for the remote association task, who gave wrong answers twice in the practice session, were excluded from the analysis. We performed between-subjects two-factor ANOVAs for each task with the number of correct responses as the dependent variable, and sample heterogeneity (online university students/YCrowd participants) and presence of incentives (contingent/fixed pay) as the independent variables. For the anagram task, all main effects and interactions were not significant (all F-values < 1.47, all p-values > 0.100). For the remote association task, all main effects and interactions were not significant (all F-values < 1.75, all p-values > 0.100). For the in-anagram task, the main effects were not significant (all F-values < 1.56, all p-values > 0.100), but the interaction was significant (F(1,195) = 4.01, partial $\eta^2$ = 0.020, p = 0.047), and the results of the simple main effect test showed that the number of correct responses decreased when the salary was fixed only for the YCrowd participants (t(195) = 3.00, d = 0.49, p = 0.003) (see Fig 2(C)). It should be noted that we also found statistical significance only for the YCrowd participants in the in-anagram task when using the Mann–Whitney U-test. In addition, the regression analysis that controlled for age, gender and order showed a similar trend (see S5–S7 Tables and S3 Fig in S1 File).

Thus, in the YCrowd participants, only the dull task resulted in lower performance without external incentives. Since no such effect was found for the university student sample, this result may be due to differences in the perceived authorities between the university student participants and the YCrowd participants.

**A post hoc power analysis.** We conducted a series of post hoc power analyses. For the group task, the effect size for the PGG contribution was large (no-punishment, d = 0.70; punishment, 0.87), but small for the amount of punishment (d = 0.34). The estimated powers were calculated with α = 0.05: the power was 0.52, 0.72, and 0.17, respectively. The power for the

PGG contribution was relatively high, but did not reach 80%. It will be necessary to revalidate the results after a prior sample size determination. For the individual tasks, the non-significant main effects and interactions had very low effect size (partial $\eta^2$ values $< 0.009$) and low estimated power (power $< 0.27$). For the interaction in the in-anagram task, which was significant, the effect size was relatively small (partial $\eta^2 = 0.202$), and the estimated power was 0.50, which was not sufficient. Future studies should also address this issue.

## General discussion

In this study, we examined the experimental situation (laboratory/online) effect on group cooperation and individual performance. We additionally collected data from the online-worker sample to examine how sample heterogeneity (students/online-worker) affected behavior.

### Group task

First, we found no significant differences for the influence of experimental situation in the PGG. Cooperative behaviors in groups with and without punishment did not differ across experimental situations. Considering the trends in previous studies that examined the effects of experimental situations [3–6], the possibility of physical distance suppressing prosocial behavior may be of little concern (with the exception of the studies of Schmelz and Ziegelmeyer [5] and Prissé and Jorrat [7]). Punishment behavior, examined for the first time in this study, was similarly unaffected by physical distance.

However, significant influences of sample heterogeneity were observed. In the PGG, the online-worker sample contributed less than the university student sample did, and the effect of punishment for facilitating cooperation was weaker. These results are interesting because previous studies have often shown the opposite tendency, with heterogeneous samples including online-workers being rather more prosocial than student samples [14–18]. Although we do not know why we found the opposite tendency, this finding has two implications. First, these results casts doubt on the universality of the finding in many previous studies that students are less prosocial than heterogeneous samples. Second, the finding that prosocial tendency varies across samples is universal, although we note that the lower incentive in YCrowd experiment might affect the results as discussed in the additional study.

### Individual task

As with the group task, a null result across experimental situations was also true for the individual tasks except the remote association task, where the presence of incentives had a negative effect on performance only in the laboratory situation. Thus, the pressure due to both the laboratory and the external incentive had a negative effect only on the more difficult and creative task. However, the interaction in the remote association task was marginally significant and the estimated power was insufficient, so we should carefully consider the replicability of these results.

In regard to the effect of sample heterogeneity, only the online-worker sample showed a decrease in performance in the dull (in-anagram) task without incentive (fixed-pay condition). This indicates that the effect of external motivation that is important in dull tasks also varied depending on the sample: the online-worker sample who are supposedly less committed to the experiments than university students need to be incentivized externally for such tasks.

### Implications and limitations

We conclude that the influence of experimental situations was small but that of the sample heterogeneity was relatively large, especially for prosocial behavior. When interpreting the results

of online experiments, which often confound the effect of the situation with that of the sample heterogeneity, it is important to put more emphasis on differences in sample heterogeneity rather than in experimental situations. These tendencies are in accordance with previous reports that collected data in Western cultures [3, 4, 6–13]. Thus, this study provides evidence of the cultural universality of these tendencies.

If we do not need to be much concerned with the effects of experimental situations, conducting online experiments using university students may be promising. This method may be actively applied in research institutes that do not have sufficient experimental facilities, because researchers can collect samples with less noise and dropout.

The finding of weak effects of the experimental situation has implications for the real world. Office managers may not have to worry much about free riding or laziness when employees change their workplace from office to home. However, in different task types such as highly challenging creative tasks, the work environment may have an influence, and further studies are necessary. Moreover, since the group of workers changes in the case of outsourcing, obtaining an appropriate worker group should be emphasized.

It is important to note to what extent our findings can be generalized. First, the laboratory and online situation in our main study might not be enough to manipulate the "social distance" because the participants in the laboratory were separated by partitions and did not have a chance to directly interact with each other. This weak manipulation would lead to weak observed effect size in the main study. However, a study that manipulated the extent of anonymity showed no such differences [3]. In any case, further research is needed. Second, the types of tasks were limited; we only conducted a few tasks (e.g., PGG, anagram, and remote association tasks), and whether similar results can be obtained with other economic games and individual tasks should be further investigated. For example, more creative tasks which need new ideas such as "unusual uses test" [50] should be examined. Third, we believe that the situations and motivations of online-workers are more varied than those of students online. Since performance would be context-dependent, we need to be careful about the generalizability of the results of the student sample in regard to the main study. Finally, the results are based on a limited samples in Japan (students at a local university and registered users of YCrowd), and it is unclear whether similar results can be obtained in different samples.

With the widespread use of online experiments in the research field and remote work in society, it is becoming increasingly important to evaluate the impacts of situation and sample heterogeneity. The accumulation of data is of utmost importance to improve the accuracy of evaluation, and the findings of our study contribute to this field.

## Supporting information

**S1 File. Supporting information.** Supplementary analysis including S1–S7 Tables and S1–S3 Figs, and supplementary method.
(DOCX)

**S1 Data.**
(XLSX)

## Acknowledgments

We would like to thank Yoshiyuki Ueda for giving us the original Japanese version and the data for the anagram task.

## Author Contributions

**Conceptualization:** Hiroki Ozono, Daisuke Nakama.

**Data curation:** Hiroki Ozono.

**Formal analysis:** Hiroki Ozono.

**Funding acquisition:** Hiroki Ozono.

**Investigation:** Hiroki Ozono.

**Methodology:** Hiroki Ozono, Daisuke Nakama.

**Project administration:** Hiroki Ozono.

**Resources:** Hiroki Ozono.

**Software:** Hiroki Ozono.

**Supervision:** Daisuke Nakama.

**Validation:** Hiroki Ozono, Daisuke Nakama.

**Visualization:** Hiroki Ozono.

**Writing – original draft:** Hiroki Ozono.

**Writing – review & editing:** Hiroki Ozono, Daisuke Nakama.

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
