## [Decision Letter · Decision Letter 0]

15 Sep 2021

PONE-D-21-25066Effects of experimental situation on group cooperation and individual performance: comparing laboratory and online experimentsPLOS ONE

Dear Dr. Ozono,

Thank you for submitting your manuscript to PLOS ONE. After careful consideration, we feel that it has merit but does not fully meet PLOS ONE’s publication criteria as it currently stands. Therefore, we invite you to submit a revised version of the manuscript that addresses the points raised during the review process.

You will find two extensive reports (one pasted, the other attached). Both reviewers provide a number of interesting comments and suggestion that you should address. Please consider that I will send the paper back to the same referees. Although they ask for a number of changes and clarifications my own interpretation is that both reviewers are fairly positive.

We look forward to receiving your revised manuscript.

Kind regards,

Pablo Brañas-Garza, PhD Economics

Academic Editor

PLOS ONE

Journal Requirements:

"This work was supported by the JSPS KAKENHI (18K13272) ."

"HO was supported by the JSPS KAKENHI (18K13272). http://www.jsps.go.jp/english/. This funder had no role in study design, data collection and analysis, decision to publish, or preparation of the manuscript."

Reviewers' comments:

Reviewer's Responses to Questions

**Comments to the Author**

1. Is the manuscript technically sound, and do the data support the conclusions?

Reviewer #1: Yes

Reviewer #2: Partly

2. Has the statistical analysis been performed appropriately and rigorously? 

Reviewer #1: Yes

Reviewer #2: Yes

3. Have the authors made all data underlying the findings in their manuscript fully available?

Reviewer #1: Yes

Reviewer #2: No

4. Is the manuscript presented in an intelligible fashion and written in standard English?

Reviewer #1: Yes

Reviewer #2: No

5. Review Comments to the Author

Reviewer #1: The manuscript is really clear, it is easy to understand the reasoning of the authors.

" mainly because of the ease of recruitment and implementation" -> I think laboratories are used because they are a controlled environment. The interest of online environment is precisely to drastically reduce the difficulty of recruitment and collection of datas.

"Social distance theory, which discusses the effect on behavior of social distance among participants," -> Explain a bit what is the effect of social distance (how it works).

"principal–agent problem" -> I do not know what it means.

"so the motive to punish may increase to maintain cooperation." -> It can also decrease since participants were not expected to cooperate, therefore punishers might interpret that there is no/less punishment to give to people who thought, similarly to them, that cooperation cannot be sustained.

" the online participants with their anonymity would be less likely to hesitate in punishing" -> Since the only difference between the two environments when talking about how participants perceive others participants is... the perceived physical presence of others... I think this factor is too light, not enough to make Lab subjects consider how they interact with others as sufficiently different than Online subjects to elicit a difference that would be significant on t-tests.

"laboratory situations where social distance is close." -> I disagree. Social distance is close when there are interactions between people, because they create inside the mind of subjects the belief of belonging to a group. I would even say that social distance can only be classified as "close" when the group has taken enough existence to have his members perform group thinking.

"They argued that the latter is caused by “choking under pressure” due to excessive incentive." -> Having made sport in my life, "choking under pressure" is relatable to a different context: when there is something to size (victory) and that the person internally collapse in front of it. This kind of situation is rather relatable to "social conformity", i.e behaving as you believe that the figure of authority wants you to behave, in order to fit with the norm of good individual. This is why you find and should expect this effect in the Lab: subjects are observed by experimenters and know it. Even see it.

"performed the PGG without punishment, followed by the PGG with punishment" -> Can you justify why you are not also doing it on the reverse side ? Because playing PGG with punishment after normal PGG could alter results in both the direction of more and less punishment. Players who saw cooperation collapse in the first PGG (the expected result) might use punishment more heavily to maintain cooperation, but they might also psychologically resign and accept the "fatality" of non-cooperation. In both cases, it means that their behavior in PGG with punishment is not "pure". And in all cases, the behavior in PGG with punishment is necessarily influenced by the perception of having made the standard PGG just before.

"For every one point used to reduce another member, this other member was reduced by two points" -> I agree that it might be too small of a punishment. 1:3 seems a better ratio to me, because you can significantly punish without losing much, meaning that you reproach bad behavior more easily.

Note: I wonder what "creative" exactly means, in his scientific definition. My definition of "creativity" is not exactly in accordance with the implicit one in the anagram task.Creativity is supposed to propose something new, which is slightly different from making associations to find an already known word.

Note #2: The tasks are well-explained.

" no particular trends worthy of report were detected" -> Interesting, because I am doing a similar experiment (Lab vs Online) and I have results associated with Questionnaires: subjects in the Lab are providing more desirable answers about themselves than Online subjects (I.e observer effect), but it seems to trigger only with questions that are somehow "salient" on social appropriateness.

"the booths" -> What is a "booth".

Note: The booth environment is halfway between "Online" and "Physical Presence" conditions. Not even talking about "Physical Presence + Social Interactions" (through computer communication). It might explained the underlying trend of results: environment are too similar, therefore the only difference is the conscious/unconscious effect on subjects of being in physical presence of others and experimenters, which is a measure of Pygmalion,Demander effects...

"Zoom was used for the questions to make the setting as similar as possible to the online situation." -> Ah yes ? Why not. Interesting.

" 1 yen per point (US$1 is approximately 100 yen)." -> Are these amounts considerable in Japan ? I think you should precise that the average payment was slightly above the average daily wage if it was the case, such that we know that subjects were playing for stakes. Because right now, I understand that the payment of starting amounts in PGG is 4$ for the whole PGG conditions. Which is not a lot from my European viewpoint...

"order of the tasks..." -> Why is it not a problem that remote association is never first ?

" basic participation fee of 500 yen." -> Perhaps precise earlier in the paper how subjects are paid, or better put their average payment.

"These results imply that the introduction of punishment had some effect in promoting and maintaining group coopereation."

-> Known result, therefore important to replicate it in the Online environment to verify his existence there.

"t the effect of punishment was very weak in Japan" -> Unclear to me. I assume hat you mean : "Punishment does not increase cooperation with Japanese students". Because you talk of Culture, a bit of explanation would make the paper more tasty. I think that what you would say would be: "Japan is an individualistic culture, and therefore Japanese students are perhaps less likely to punish subjects who refuse to cooperate because their social codes accept the refusal to engage in group behavior".

"Funaki points out that (...) perverse punishment -> Unclear. It seems like Funkaki talk about Japanse students, but why ? Because Japanese students would actually punish the person who is prosocial ? Being prosocial would be considered as a norm violation in social interactions ? It is interesting to discuss.

"the higher pressure induced by the other participants and the experimenter in the laboratory" -> I think the effect is created by experimenters and the laboratory in itself, not others subjects.

"nervousness may have strongly affected performance in the remote association task"-> Social pressure might inhibit creative behavior due to the perception of being judged by an authority figure.

"Another possibility is that the participants might be highly motivated to perform the task seriously" -> I agree. Therefore, it is arguable that because of motivated samples and close experimental condition (cf.above why I think that), the no-result was to expect on the overall trend of results.

Ycrowd users are also likely to be motivated, since they make the effort of registering and connecting to a website to perform tasks. Additionally, they might do that for a living, which would make them even more motivated than students to perform. I do not believe that this sample resolve the motivation issue.

"Specifically, the participation fee was set at 30 yen, and the PGG rate was set at 0.2 yen per point" -> I see an issue in not having the same experimental payments. If you have different results than Lab and Online with students, it can be argued as being the cause. If you have a no-result, then it would rather suggest that the scale of payment is not important, in accordance with previous papers who showed that for M-Turk scale of payment is not importatn.

I quote my own Literature Review on this topic : "Mason and Watts (2009) and Marge, Banerjee and Rudnicky (2010) showed that AMT wages influenced the quantity but not the quality of work which suggest that payment scale is not an issue, in accordance with the meta-analysis of Camerer and Hogarth (1999)".

"Mann–Whitney U-test showed that the YCrowd participants contributed less than the university student participants in the PGG without punishment (U = 137.5, Z = 2.260, p = 0.024) and with punishment (U = 111.5, Z = 2.891, p = 0.004)."

-> Ok, so we fall on my previous comment.

If I have followed, the "standard" endowment in Lab/Online "PGG + PGG with punishment" is equivalent to 1point = 1 Yen, i.e 400 Yen = 4$. Now, with 1 point = 0.2 yen, it means that subjects gain 80 Yen = 0.8$. So perhaps what is happening is that the endowment of subjects is perceived as small and they are therefore more "conservative" to not lose the few that they have been able to acquire.

Important: It is not an interpretation "on the flight" (French: "au vol"). Without the context of the value of a Yen, it does not pretend at truthfulness.

"Groups that had more than one case of exceeding the time limit were excluded from the analysis." -> Seems coherent to me, since Online environment can trigger loss of attention, and until it is precisely known to which proportion it affects results, strict selection of results is a better choice. (Note: In my experiment, I have few outliers in Time. But I suspect that slowest subjects in the Online condition are a bit distracted by Internet notifications).

'The reason for this may be a cultural characteristic of Japan," -> Perhaps it is only Internet samples that have less expectations of cooperations, since this kind of subjects are more likely to be isolated in their lives, while students (whether Lab or Online) have the feeling of belonging to something (the university). Also, if they know they are with similar peers, it might triggers prosociality by recognition of similarity. This is, said otherwise, "social distance". Larger social distance in the Online environment between participants create less prosociality than closer social distance between students. Additionally, students might mutually know that they are in a precarious/vulnerable social status and therefore be inclined to help each others.

"leaving our results open to various interpretations" -> Indeed.

"and population (online university students/YCrowd participants) and presence of incentives (contingent/fixed pay) as the independent variables." -> (0=, 1=) , (0=, 1=) so that we know the meaning behind the interaction effect more intuitively. Also, I do not remember if you did the same parenthesis in the first experiment. I think not.

" but the interaction was significant " -> Perhaps put the number, so that we intuitively know the direction of the effect.

" to differences in sincerity of the attitude" -> I would rather say "intrinsic motivation". This general online sample is intrisically less motivated and therefore will not perform annoying tasks. But he will normally perform standard tasks that are "intellectually stimulating" to him.

"he possibility of physical distance" -> I think that "physical distance x social contact" would have an effect, but not physical distance alone.

Overall Conclusion: The manuscript is clear, well-interpreted. The experimental design is simple and to the point. Results do what they are supposed to do and make a contribution to the literature. It is good for me.

Additional Comment: I am currently finishing to writte a paper on "Lab vs Online". My subjects were exclusively students, and they played : Convex Time Budget, Multiple Price Lists, Holt-Laury, Dictator Game (Charity donation version), CRT, Numeracy questionnaire, Questionnaire on various charateristics. I similarly find a no-result overall, but similarly to you I occasionally find small differences in details. My interpretation is that these difference arise because of Pygmalion of Demander effect, depending on the context we are talking about.

I suspect that the kind of task eliciting this kind of reactions are "salient" questions for demander effect and "sensible" experimental trials for Pygmalion effect. "salient" means that the question is very precise on the behavior of the subjects and the subject can instantly grasp if it is good/bad behavior, "sensible" means trials in CTB (or HL) that are "ambiguous" in the answer of subjects, compared to others more straightforward trials. Lab subjects aim at larger amounts, i.e "increased productivity" in terms of Pygmalion effect if they were factory workers.

I think my advisor would have no issue sharing the paper.

Reviewer #2: The research question of the paper is interesting but a more detailed analysis of the data is required. The structure of the analysis and sections of the paper needs to be revised in order to make the results more legible.

I attached a document with my comments.

6. PLOS authors have the option to publish the peer review history of their article (what does this mean?). If published, this will include your full peer review and any attached files.

Reviewer #1: **Yes: **Benjamin Prissé

Reviewer #2: No

---

## [Author Response · Author response to Decision Letter 0]

30 Nov 2021

*We greatly appreciate the comments and careful review of our paper by the editor and two reviewers. We have revised the paper taking into account the reviewers’ helpful suggestions and comments. We now address each comment made by the editor and two reviewers. Our responses are marked with asterisks.

[Reply to the Editor]

Editor’s Comment:

Thank you for submitting your manuscript to PLOS ONE. After careful consideration, we feel that it has merit but does not fully meet PLOS ONE’s publication criteria as it currently stands. Therefore, we invite you to submit a revised version of the manuscript that addresses the points raised during the review process.

You will find two extensive reports (one pasted, the other attached). Both reviewers provide a number of interesting comments and suggestion that you should address. Please consider that I will send the paper back to the same referees. Although they ask for a number of changes and clarifications my own interpretation is that both reviewers are fairly positive.

We look forward to receiving your revised manuscript.

Kind regards,

Pablo Brañas-Garza, PhD Economics

Academic Editor

PLOS ONE

Reply: 

*We thank you for giving us the opportunity to revise our manuscript. We made a point-by-point response to all issues raised by the reviewers.

[Reply to the Journal Requirements:]

Journal Requirements:

Reply: 

*We collected the styles according to the guidelines. I appreciate your suggestion.

"This work was supported by the JSPS KAKENHI (18K13272) ."

"HO was supported by the JSPS KAKENHI (18K13272). http://www.jsps.go.jp/english/. This funder had no role in study design, data collection and analysis, decision to publish, or preparation of the manuscript."

Reply: 

*We deleted the funding information from the acknowledgement. We do not need to change the Funding statement. We replied with the same comment in the cover letter.

[Reply to the Reviewer 1]

Reviewer 1’s Comment (1):

The manuscript is really clear, it is easy to understand the reasoning of the authors.

" mainly because of the ease of recruitment and implementation" -> I think laboratories are used because they are a controlled environment. The interest of online environment is precisely to drastically reduce the difficulty of recruitment and collection of datas.

Reply: 

* We agree with you. We changed the merit of laboratories to “the ease of providing a controlled environment” (page 3 line 27 to 28).

Reviewer 1’s Comment (2):

"Social distance theory, which discusses the effect on behavior of social distance among participants," -> Explain a bit what is the effect of social distance (how it works).

Reply: 

* We added the explanation and a reference (page 5 line 65 to 70).

Reviewer 1’s Comment (3):

"principal–agent problem" -> I do not know what it means.

Reply: 

* We added a brief explanation and a reference (page 6 line 80 to 83).

Reviewer 1’s Comment (4):

"so the motive to punish may increase to maintain cooperation." -> It can also decrease since participants were not expected to cooperate, therefore punishers might interpret that there is no/less punishment to give to people who thought, similarly to them, that cooperation cannot be sustained.

Reply: 

* We totally agree with your opinion. We deleted this sentence because it does not work as reasoning.

Reviewer 1’s Comment (5):

" the online participants with their anonymity would be less likely to hesitate in punishing" -> Since the only difference between the two environments when talking about how participants perceive others participants is... the perceived physical presence of others... I think this factor is too light, not enough to make Lab subjects consider how they interact with others as sufficiently different than Online subjects to elicit a difference that would be significant on t-tests.

Reply: 

* We agree with you. We discuss this issue in the limitations section of the general discussion (page 38 line 652 to 658).

Reviewer 1’s Comment (6):

"laboratory situations where social distance is close." -> I disagree. Social distance is close when there are interactions between people, because they create inside the mind of subjects the belief of belonging to a group. I would even say that social distance can only be classified as "close" when the group has taken enough existence to have his members perform group thinking.

Reply: 

*We changed the word to “physical distance” (page 7 line 98) and we also discuss this issue in the limitations section of the general discussion (page 38 line 652 to 658).

Reviewer 1’s Comment (7):

"They argued that the latter is caused by “choking under pressure” due to excessive incentive." -> Having made sport in my life, "choking under pressure" is relatable to a different context: when there is something to size (victory) and that the person internally collapse in front of it. This kind of situation is rather relatable to "social conformity", i.e behaving as you believe that the figure of authority wants you to behave, in order to fit with the norm of good individual. This is why you find and should expect this effect in the Lab: subjects are observed by experimenters and know it. Even see it.

Reply: 

* In our context, “pressure” includes social pressure, which is elicited by an experimenter and other participants. To clarify this point, we add ”which leads to higher social pressure” in the latter part (page 8 line 117). Reviewer 1 emphasizes the effect of experimenters rather than other participants. This might be possible, but it is difficult to make a conclusion with our experimental settings and results. Thus, we clearly mention this inconclusiveness in the discussion (page 25 line 421 to 423).

Reviewer 1’s Comment (8):

"performed the PGG without punishment, followed by the PGG with punishment" -> Can you justify why you are not also doing it on the reverse side ? Because playing PGG with punishment after normal PGG could alter results in both the direction of more and less punishment. Players who saw cooperation collapse in the first PGG (the expected result) might use punishment more heavily to maintain cooperation, but they might also psychologically resign and accept the "fatality" of non-cooperation. In both cases, it means that their behavior in PGG with punishment is not "pure". And in all cases, the behavior in PGG with punishment is necessarily influenced by the perception of having made the standard PGG just before.

Reply: 

* We totally agree with you. It is better to counterbalance the order of the PGG with and without punishment. But we did not expect to obtain enough samples and previous studies sometimes did not counterbalance the order (Arecher & Gacther, 2018), so we did not attempt to counterbalance the order. We mentioned this in the methods section (page 10 line 155 to 158).

Reviewer 1’s Comment (9):

"For every one point used to reduce another member, this other member was reduced by two points" -> I agree that it might be too small of a punishment. 1:3 seems a better ratio to me, because you can significantly punish without losing much, meaning that you reproach bad behavior more easily.

Reply: 

* We totally agree with you. We have already mentioned this point in the discussion, in which “Although we established this parameter to avoid a ceiling effect of contribution in PGG due to excessive power of the punishment, this low efficiency might weaken the punishment effect more than we expected.” We think this is sufficient.

Reviewer 1’s Comment (10):

Note: I wonder what "creative" exactly means, in his scientific definition. My definition of "creativity" is not exactly in accordance with the implicit one in the anagram task. Creativity is supposed to propose something new, which is slightly different from making associations to find an already known word.

Reply: 

* As you said, it was problematic to consider the anagram task as a creative task. We could only find one Japanese paper that clearly argued the anagram task as a creative task. On the other hand, the remote association task is clearly considered as a creative task. Therefore, we decided to position the anagram task as a task that requires convergent thinking and changed the wording throughout the paper. Especially, we add the explanation for the tasks (page 8 line 128 to 133) and the discussion for anagram task (page 24 line 410 to 413)

Reviewer 1’s Comment (11):

Note #2: The tasks are well-explained.

" no particular trends worthy of report were detected" -> Interesting, because I am doing a similar experiment (Lab vs Online) and I have results associated with Questionnaires: subjects in the Lab are providing more desirable answers about themselves than Online subjects (I.e observer effect), but it seems to trigger only with questions that are somehow "salient" on social appropriateness.

Reply: 

* We appreciate your suggestions. We reported the data of demographic factors and measured characteristics in the supplementary analysis 1(see S1 file).

Reviewer 1’s Comment (12):

"the booths" -> What is a "booth".

Note: The booth environment is halfway between "Online" and "Physical Presence" conditions. Not even talking about "Physical Presence + Social Interactions" (through computer communication). It might explained the underlying trend of results: environment are too similar, therefore the only difference is the conscious/unconscious effect on subjects of being in physical presence of others and experimenters, which is a measure of Pygmalion,　Demander effects...

"Zoom was used for the questions to make the setting as similar as possible to the online situation." -> Ah yes ? Why not. Interesting.

Reply: 

* For better understanding, we attached a video (page 15 line 250). In addition, we discuss the weak manipulation of social distance in the limitation section of the general discussion (page 38 line 652 to 658). 

Reviewer 1’s Comment (13):

" 1 yen per point (US$1 is approximately 100 yen)." -> Are these amounts considerable in Japan ? I think you should precise that the average payment was slightly above the average daily wage if it was the case, such that we know that subjects were playing for stakes. Because right now, I understand that the payment of starting amounts in PGG is 4$ for the whole PGG conditions. Which is not a lot from my European viewpoint...

Reply: 

*We added sentences that mention how to pay and the average payment in the participants section. In addition, we added the average time to complete the session, so that readers can understand that the payment is appropriate(page 11 line 172 to 178).

Reviewer 1’s Comment (14):

"order of the tasks..." -> Why is it not a problem that remote association is never first ?

Reply: 

* We agree that this is a problem, so we set the remote association first in the additional study. We address this issue in the section of additional study (page 30 line 507 to510).

Reviewer 1’s Comment (15):

" basic participation fee of 500 yen." -> Perhaps precise earlier in the paper how subjects are paid, or better put their average payment.

Reply: 

* We have already addressed this issue in Reviewer 1’s comment (13).

Reviewer 1’s Comment (16):

"These results imply that the introduction of punishment had some effect in promoting and maintaining group coopereation."

-> Known result, therefore important to replicate it in the Online environment to verify his existence there.

Reply: 

* We emphasized that we also found the effects of punishment online (page 22 line 362).

Reviewer 1’s Comment (17):

"t the effect of punishment was very weak in Japan" -> Unclear to me. I assume hat you mean : "Punishment does not increase cooperation with Japanese students". Because you talk of Culture, a bit of explanation would make the paper more tasty. I think that what you would say would be: "Japan is an individualistic culture, and therefore Japanese students are perhaps less likely to punish subjects who refuse to cooperate because their social codes accept the refusal to engage in group behavior".

Reply: 

* We do not wish to discuss the cultural differences more deeply because (1) this issue is not our central question and we did not collect data for other cultures, and (2) many interpretations would be possible, so we believe that we should simply present the data, not make judgmental arguments.

Reviewer 1’s Comment (18):

"Funaki points out that (...) perverse punishment -> Unclear. It seems like Funkaki talk about Japanese students, but why ? Because Japanese students would actually punish the person who is prosocial ? Being prosocial would be considered as a norm violation in social interactions ? It is interesting to discuss.

Reply: 

*Funaki did not discuss the reason for strong perverse punishment in Japan, and we do not want to make judgmental arguments. Thus, we simply said “the reason is still unclear.” (page 22 line 374 to 375).

Reviewer 1’s Comment (19):

"the higher pressure induced by the other participants and the experimenter in the laboratory" -> I think the effect is created by experimenters and the laboratory in itself, not others subjects.

Reply: 

* As we said in the reply to Reviewer 1’s comment (7), this might be possible, but is difficult to conclude. We mention this point in the end of this paragraph (page 25 line 421 to 423).

Reviewer 1’s Comment (20):

"nervousness may have strongly affected performance in the remote association task"-> Social pressure might inhibit creative behavior due to the perception of being judged by an authority figure.

Reply: 

* We changed the word “nervousness” to “social pressure” to clarify our point (page 25 line 416).

Reviewer 1’s Comment (21):

"Another possibility is that the participants might be highly motivated to perform the task seriously" -> I agree. Therefore, it is arguable that because of motivated samples and close experimental condition (cf.above why I think that), the no-result was to expect on the overall trend of results.

Ycrowd users are also likely to be motivated, since they make the effort of registering and connecting to a website to perform tasks. Additionally, they might do that for a living, which would make them even more motivated than students to perform. I do not believe that this sample resolve the motivation issue.

Reply: 

*We agree with you. We wrote in a misleading way and have corrected this in the end of this paragraph (page 26 line 440 to 441).

Reviewer 1’s Comment (22):

"Specifically, the participation fee was set at 30 yen, and the PGG rate was set at 0.2 yen per point" -> I see an issue in not having the same experimental payments. If you have different results than Lab and Online with students, it can be argued as being the cause. If you have a no-result, then it would rather suggest that the scale of payment is not important, in accordance with previous papers who showed that for M-Turk scale of payment is not importatn.

I quote my own Literature Review on this topic : "Mason and Watts (2009) and Marge, Banerjee and Rudnicky (2010) showed that AMT wages influenced the quantity but not the quality of work which suggest that payment scale is not an issue, in accordance with the meta-analysis of Camerer and Hogarth (1999)".

Reply: 

* We added discussion about the possibility that the lower incentive might affect the results for the YCrowd experiment in the end of the result and discussion for group task (page 32 line 548 to 554).

Reviewer 1’s Comment (23):

"Mann–Whitney U-test showed that the YCrowd participants contributed less than the university student participants in the PGG without punishment (U = 137.5, Z = 2.260, p = 0.024) and with punishment (U = 111.5, Z = 2.891, p = 0.004)."

-> Ok, so we fall on my previous comment.

If I have followed, the "standard" endowment in Lab/Online "PGG + PGG with punishment" is equivalent to 1point = 1 Yen, i.e 400 Yen = 4$. Now, with 1 point = 0.2 yen, it means that subjects gain 80 Yen = 0.8$. So perhaps what is happening is that the endowment of subjects is perceived as small and they are therefore more "conservative" to not lose the few that they have been able to acquire.

Important: It is not an interpretation "on the flight" (French: "au vol"). Without the context of the value of a Yen, it does not pretend at truthfulness.

Reply: 

* In order to help readers understand how the value of currency was perceived by the participants in this study, we added the sentences (page 11 line 172 to 178). As for the issue of the lower incentive in YCrowd sample, we have already addressed this issue in Reviewer 1’s comment (22).

Reviewer 1’s Comment (24):

"Groups that had more than one case of exceeding the time limit were excluded from the analysis." -> Seems coherent to me, since Online environment can trigger loss of attention, and until it is precisely known to which proportion it affects results, strict selection of results is a better choice. (Note: In my experiment, I have few outliers in Time. But I suspect that slowest subjects in the Online condition are a bit distracted by Internet notifications).

Reply: 

*We totally agree with you. We decided that no specific modifications were needed here.

Reviewer 1’s Comment (25):

'The reason for this may be a cultural characteristic of Japan," -> Perhaps it is only Internet samples that have less expectations of cooperations, since this kind of subjects are more likely to be isolated in their lives, while students (whether Lab or Online) have the feeling of belonging to something (the university). Also, if they know they are with similar peers, it might triggers prosociality by recognition of similarity. This is, said otherwise, "social distance". Larger social distance in the Online environment between participants create less prosociality than closer social distance between students. Additionally, students might mutually know that they are in a precarious/vulnerable social status and therefore be inclined to help each others.

Reply: 

* We agree that the university students feel closer “social distance” than general online samples. However, as we mentioned in the main manuscript, many previous studies suggested that a general sample was more cooperative than a university student sample, which contradicts the argument and it is difficult to determine why only our student sample would feel closer social distance. In order to avoid confusing discussions, we would like to leave this as it is without any amendments.

Reviewer 1’s Comment (26):

"leaving our results open to various interpretations" -> Indeed.

"and population (online university students/YCrowd participants) and presence of incentives (contingent/fixed pay) as the independent variables." -> (0=, 1=) , (0=, 1=) so that we know the meaning behind the interaction effect more intuitively. Also, I do not remember if you did the same parenthesis in the first experiment. I think not.

" but the interaction was significant " -> Perhaps put the number, so that we intuitively know the direction of the effect.

Reply: 

* We interpreted that you recommend an analysis using a general linear model. We know that the results were consistent with the ANOVA, but some readers, especially in the psychology field, will not be familiar with general linear models. Thus, we decided not to correct this part. Instead, for more intuitive understanding, we indicate which figure shows the interaction pattern (page 33 line 569). We collected the interaction results in the main study in the same manner(page 24 line 398).

Reviewer 1’s Comment (27):

" to differences in sincerity of the attitude" -> I would rather say "intrinsic motivation". This general online sample is intrisically less motivated and therefore will not perform annoying tasks. But he will normally perform standard tasks that are "intellectually stimulating" to him.

Reply: 

*“sincerity of the attitude” is ambiguous and misleading, so we changed it to “the perceived authorities.” (page 33 line 576 to 577)

Reviewer 1’s Comment (28):

"he possibility of physical distance" -> I think that "physical distance x social contact" would have an effect, but not physical distance alone.

Reply: 

*We address the weak manipulation of “social distance” in the section of limitation of the general discussion (page 38 line 652 to 658).

Reviewer 1’s Comment (29):

Overall Conclusion: The manuscript is clear, well-interpreted. The experimental design is simple and to the point. Results do what they are supposed to do and make a contribution to the literature. It is good for me.

Additional Comment: I am currently finishing to writte a paper on "Lab vs Online". My subjects were exclusively students, and they played : Convex Time Budget, Multiple Price Lists, Holt-Laury, Dictator Game (Charity donation version), CRT, Numeracy questionnaire, Questionnaire on various charateristics. I similarly find a no-result overall, but similarly to you I occasionally find small differences in details. My interpretation is that these difference arise because of Pygmalion of Demander effect, depending on the context we are talking about.

I suspect that the kind of task eliciting this kind of reactions are "salient" questions for demander effect and "sensible" experimental trials for Pygmalion effect. "salient" means that the question is very precise on the behavior of the subjects and the subject can instantly grasp if it is good/bad behavior, "sensible" means trials in CTB (or HL) that are "ambiguous" in the answer of subjects, compared to others more straightforward trials. Lab subjects aim at larger amounts, i.e "increased productivity" in terms of Pygmalion effect if they were factory workers.

I think my advisor would have no issue sharing the paper.

Reply: 

* The argument by Reviewer 1 is very interesting, but it is difficult to make any conclusion using our experimental settings and results, so we decided not to discuss this any deeper. We appreciate you giving us your working paper. We referred to the paper as one of the studies examining the situational differences (laboratory/online).

[Reply to the Reviewer 2]

Reviewer 2’s Comment (1):

Reviewer #2: The research question of the paper is interesting but a more detailed analysis of the data is required. The structure of the analysis and sections of the paper needs to be revised in order to make the results more legible.

The paper examined how experimental situation, more precisely, online or lab setting affects group cooperation and other individual tasks that measure performance.

In the first experiment, the author assigned students randomly to one of the treatments. In this experiment, it is plausible to measure the effect of online setting compared to lab setting.

Also, an online experiment using Yahoo crowd was made (Ycrowd). But this experiment involves another type of sample (with different characteristics).

Results from experiment 1 suggest no difference in cooperation, while comparing Ycrowd with Lab setting the results show differences.

General comments:

1. The random assignment makes the two treatments (online or lab setting) comparable, and suggest internal validity. This assures that author could find the causal effect of the treatment.

2. However, in experiment Ycrowd this is not truth. These subjects, probably are not comparable, and even if they were, the non-observable characteristics of this group would be different compare to lab students. Saying this, It is not clear for me how this experiment would add more information about the topic (which is very interesting for me and I agree with the author that this kind of research is necessary in the literature). I suggest the following comments:

• First, author need to show that in experiment 1 both treatments were balance in different observable characteristics (mean tests and p values shown in a table).

Reply: 

* We appreciate your suggestions. We reported the data and the statistical test for demographic factors and measured characteristics in the supplementary analysis 1(see S1 Table in S1 File) and referred to it in the main text (page 11 line 171 to 172). We found no difference between laboratory and online experiment for student sample.

Reviewer 2’s Comment (2):

• For the second comment, author need to show that both samples (Ycrowd and Lab) are comparable. A paper that analyzed this is Brañas-Garza et al. (2013) “Experimental subjects are not different”, I suggest to add it in the introduction.

Reply: 

*We added sentences explaining the importance of comparing samples by referring to Exadaktylos et al. (page 9 line 140 to 149).

Reviewer 2’s Comment (3):

• Third, the null effects in the first experiment could be explained it because a lack of statistical power calculations. I suggest to run power calculation in other to see if it is possible to have significant results with a higher number of observations.

Reply: 

*Indeed, it is important to report the power. We appreciate your suggestion. We added a section for a post hoc power analysis both in the main study (page 26 line 443 to page 27 line 465) and the additional study (page 34 line 579 to 590).

Reviewer 2’s Comment (4):

• Fourth, author need to add tables with regression results controlling for age, gender and others controls.

Reply: 

*We added the tables with regression analysis in the supplementary analysis(see S2-7 Tables in S1 File) . These results showed the similar tendency after controlling for demographic factors and order of the individual tasks. We found the order effect only for the inverse-anagram task in the YCrowd sample, so we excluded the sentence “We found no order effects, so the data were combined in the analysis” and reported that we found the same trends after controlling for order (page 24 line 402 to 403 and page 33 line 571 to 573).

Reviewer 2’s Comment (5):

Specific comments:

1. The English needs to be revised:

a. For example, in the first phrase (line 25) appears “perform something”, which sounds wear.

b. Also lines 31 to 333.

c. What does “behavioral tendencies” means? I thinks it refers to standard behaviour or even could be results (line 33).

Reply: 

* We changed the word “perform something” to “perform task.” Also, we changed the word “behavioral tendencies” to “behavior.”

Reviewer 2’s Comment (6):

d. A more appropriate terms is heterogenous sample than general sample in line 31. Also could be non-standard experimental subjects.

e. Line 37, again general sample online. Also it is not the effect of the population, it is the effect of the sample heterogeneity.

f. I would not use the term population, because these type of subjects (Mturkers or others) are not representative of the whole population.

Reply: 

*We changed the word “general sample” to “heterogeneous sample” or “online-worker sample” according to the context. Also, we changed the word “population” to “sample heterogeneity”, “sample” or “sample group.”

Reviewer 2’s Comment (7):

g. Line 8: there is a typo is Arechar.

Reply: 

*We corrected this.

Reviewer 2’s Comment (8):

2. The structure does not follows the guideline instructions. It is too long and needs to follows the structure of Plos One´s papers.

Reply: 

*We appreciate your suggestion. We corrected the styles according to the guidelines of PLoS ONE. The guideline says “Manuscripts can be any length. There are no restrictions on word count, number of figures, or amount of supporting information. We encourage you to present and discuss your findings concisely.” We tried to shorten the manuscript by eliminating the “General Discussion” section which simply repeats the summary of our study (-34 words) and mention about the previous studies in the Research Question 1 section (-31 words). In addition, we tried to be as succinct and to the point as possible throughout the manuscript. Still, in order to respond to insightful comments given by all of the reviewers, we had no choice but to add some sentences which increased final word count of the manuscript. I hope our effort in making this revision concise meets your approval.

---

## [Decision Letter · Decision Letter 1]

28 Feb 2022

PONE-D-21-25066R1Effects of experimental situation on group cooperation and individual performance: comparing laboratory and online experimentsPLOS ONE

Dear Dr. Ozono,

Thank you for submitting your manuscript to PLOS ONE. After careful consideration, we feel that it has merit but does not fully meet PLOS ONE’s publication criteria as it currently stands. Therefore, we invite you to submit a revised version of the manuscript that addresses the points raised during the review process.

I consider that your paper address now the previous concerns but, as one referee suggests, there are some considerations that could improve the presentation of the findings. So, please consider how to include the suggestions done by Reviewer 3. They are minor changes that could be easily used to improve the paper.

We look forward to receiving your revised manuscript.

Kind regards,

Alfonso Rosa Garcia

Academic Editor

PLOS ONE

Journal Requirements:

Reviewers' comments:

Reviewer's Responses to Questions

**Comments to the Author**

1. If the authors have adequately addressed your comments raised in a previous round of review and you feel that this manuscript is now acceptable for publication, you may indicate that here to bypass the “Comments to the Author” section, enter your conflict of interest statement in the “Confidential to Editor” section, and submit your "Accept" recommendation.

Reviewer #1: All comments have been addressed

Reviewer #3: All comments have been addressed

2. Is the manuscript technically sound, and do the data support the conclusions?

Reviewer #1: Yes

Reviewer #3: Yes

3. Has the statistical analysis been performed appropriately and rigorously? 

Reviewer #1: Yes

Reviewer #3: Yes

4. Have the authors made all data underlying the findings in their manuscript fully available?

Reviewer #1: Yes

Reviewer #3: Yes

5. Is the manuscript presented in an intelligible fashion and written in standard English?

Reviewer #1: Yes

Reviewer #3: Yes

6. Review Comments to the Author

Reviewer #1: You have addressed the comments made in a well-made fashion, therefore I accept the paper. I thank you for quoting our paper in the new version of this paper.

Reviewer #3: The authors have addressed the reviewers’ comments adequately, in my opinion. This piece offers new interesting insights to a growing literature. Thus, the 11 comments below concentrate on the presentation of the findings, rather than on their fundamentals. I think such comments can be addressed relatively quickly, if the authors find them useful.

1. I believe the readership could assimilate your design more easily in the context of an explicit 2x2x2 environment [experimental situation x incentive scheme x sample heterogeneity] where you have the first four combinations covered [experimental situation x incentive scheme] in the first two research questions. Then, you could present a case for the third set of combinations [sample heterogeneity], where you focus only on online situations – e.g. it is fine to have the most elusive combinations [lab & heterogeneous] outside the scope of this type of paper because of the logistical and practical insights obtained. This approach would also allow you to integrate the “Additional Experiment” section more organically, potentially as a third research question.

2. Page 5, line 66. It is not clear to me why social distance theory predicts substantially different (prosocial) behavior. The way it is explained seems quite in line with the idea suggested by the observer effect (i.e. that prosociality would flourish with physical proximity, including the experimenter).

3. Page 6, line 79. Add “, that participants online tend to be less prosocial”.

4. Page 6 line 84. I’d argue that it also warrants a more nuanced comparison of the results because the papers discussed, although focused on experimental situations, offer insights based on samples from different countries. Therefore, to the extent that there is an interaction between situations and sample compositions, pooling their results is not ideal.

5. Page 7 line 91. I believe they found the online sample, which was older, to be more cooperative – even after controlling for demographics / overall sample heterogeneity.

6. Research question 2. This question could emphasize a bit more that the comparisons rely on student samples. Ideally, you could also highlight that the hypotheses relative to performance are quite context-dependent: it is very different to compare a sample of students in the lab and online versus, for example, crowds that opt to take surveys out of boredom while commuting or another group that takes surveys for a living and concentrates in front of their computers.

7. Page 10 line 156. This argument could be strengthened by citing Fehr & Gächter (2002), who found no evidence of order effects.

8. Page 11 line 179. I think this line should contextualize the prevalence of a pandemic and the potential effect on who decides to go to a lab.

9. Page 17 line 286. It would be informative to provide brief statistics about how many groups were excluded here, and how many participants timed out (none?). A line justifying why full contributions were set as the default (and the inclusion of an automatic program) would be useful too.

10. Page 19 line 309. It would be useful to know more about such a preliminary experiment.

11. Page 21 line 345. Results from the Yahoo crowdsourcing sample currently come a bit out of the blue. I think this sample should be contextualized a bit more before it is introduced here (e.g. by integrating the study as suggested in point 1), and in the Figures. Other than the abstract, no mention of YCrowd is made until this point.

7. PLOS authors have the option to publish the peer review history of their article (what does this mean?). If published, this will include your full peer review and any attached files.

Reviewer #1: **Yes: **Benjamin Prissé

Reviewer #3: No

---

## [Author Response · Author response to Decision Letter 1]

31 Mar 2022

*We greatly appreciate the comments and careful review of our paper by the editor and two reviewers. We have revised the paper taking into account the reviewers’ helpful suggestions and comments. We now address each comment made by the editor and two reviewers. Our responses are marked with asterisks.

[Reply to the Editor]

Editor’s Comment:

Dear Dr. Ozono,

Thank you for submitting your manuscript to PLOS ONE. After careful consideration, we feel that it has merit but does not fully meet PLOS ONE’s publication criteria as it currently stands. Therefore, we invite you to submit a revised version of the manuscript that addresses the points raised during the review process.

I consider that your paper address now the previous concerns but, as one referee suggests, there are some considerations that could improve the presentation of the findings. So, please consider how to include the suggestions done by Reviewer 3. They are minor changes that could be easily used to improve the paper.

We look forward to receiving your revised manuscript.

Kind regards,

Alfonso Rosa Garcia

Academic Editor

PLOS ONE

Reply: 

*We thank you for giving us the opportunity to revise our manuscript. We made a point-by-point response to all issues raised by the reviewers.

[Reply to the Journal Requirements:]

Journal Requirements:

Reply: 

* We appreciate your suggestion. We do not refer to retracted papers.

[Reply to the Reviewer 1]

Reviewer 1’s Comment (1):

You have addressed the comments made in a well-made fashion, therefore I accept the paper. I thank you for quoting our paper in the new version of this paper.

Reply: 

* We appreciate the important comments by the reviewer concerning the first version of our paper.

[Reply to the Reviewer 3]

Reviewer 3’s Comment (1):

Reviewer #3: The authors have addressed the reviewers’ comments adequately, in my opinion. This piece offers new interesting insights to a growing literature. Thus, the 11 comments below concentrate on the presentation of the findings, rather than on their fundamentals. I think such comments can be addressed relatively quickly, if the authors find them useful.

1. I believe the readership could assimilate your design more easily in the context of an explicit 2x2x2 environment [experimental situation x incentive scheme x sample heterogeneity] where you have the first four combinations covered [experimental situation x incentive scheme] in the first two research questions. Then, you could present a case for the third set of combinations [sample heterogeneity], where you focus only on online situations – e.g. it is fine to have the most elusive combinations [lab & heterogeneous] outside the scope of this type of paper because of the logistical and practical insights obtained. This approach would also allow you to integrate the “Additional Experiment” section more organically, potentially as a third research question.

Reply: 

* We appreciate your suggestion. Indeed, this research frame (2×2×2) would be more integrative. However, we did not have such a 2×2×2 frame at the start of the study, and the online-worker sample was actually collected "additionally". In recent years, researchers have argued that the research frame should be reported as it is without changing it a posteriori (Editorial, 2020, Nature Human Behavior), and we would like to follow this policy. Therefore, we decided to just mention that we did not obtain data for the online-worker sample with the laboratory situation and the reasons for this in the Introduction (page 10 line 159 to 163).

.

Reviewer 3’s Comment (2):

2. Page 5, line 66. It is not clear to me why social distance theory predicts substantially different (prosocial) behavior. The way it is explained seems quite in line with the idea suggested by the observer effect (i.e. that prosociality would flourish with physical proximity, including the experimenter).

Reply: 

* Indeed, these two theories are conceptually similar. In this study, we do not focus specifically on the differences between the two theories, but simply trace the discussions in previous studies that compared the laboratory situation with the online situation. We added sentences to clarify our thoughts (page 5 line 70 to 73).

Reviewer 3’s Comment (3):

3. Page 6, line 79. Add “, that participants online tend to be less prosocial”.

Reply: 

* We appreciate your suggestion. We added the sentence (page 6 line 82).

Reviewer 3’s Comment (4):

4. Page 6 line 84. I’d argue that it also warrants a more nuanced comparison of the results because the papers discussed, although focused on experimental situations, offer insights based on samples from different countries. Therefore, to the extent that there is an interaction between situations and sample compositions, pooling their results is not ideal.

Reply: 

* We appreciate your suggestion. As you stated, these inconsistent results might be due to differences in sample populations, and it might not be appropriate to simply pool them together. However, all the experiments we referred to were conducted with university students in Western countries, and even in the same country (Germany), with inconsistent results (Bader et al., 2019; Schmelz & Ziegelmever, 2020). It is difficult to obtain other detailed information regarding the sample groups, so we cannot discuss this point further. We added this argument in the Introduction (page 6 line 87 to 92).

Reviewer 3’s Comment (5):

5. Page 7 line 91. I believe they found the online sample, which was older, to be more cooperative – even after controlling for demographics / overall sample heterogeneity.

Reply: 

* Indeed, they found that older participants were more cooperative even after controlling for other demographic factors. However, they found this in the no-punishment condition (see Table 6 of Arechor et al., 2018) and we do not think it is necessary to refer to it in this context. 

Reviewer 3’s Comment (6):

6. Research question 2. This question could emphasize a bit more that the comparisons rely on student samples. Ideally, you could also highlight that the hypotheses relative to performance are quite context-dependent: it is very different to compare a sample of students in the lab and online versus, for example, crowds that opt to take surveys out of boredom while commuting or another group that takes surveys for a living and concentrates in front of their computers.

Reply: 

* We appreciate your suggestion. We added the sentences in the limitation (page 39 line 685 to 688).

Reviewer 3’s Comment (7):

7. Page 10 line 156. This argument could be strengthened by citing Fehr & Gächter (2002), who found no evidence of order effects.

Reply: 

* We appreciate your advice. We added the reference to this study (page 11 line 172 to 173).

Reviewer 3’s Comment (8):

8. Page 11 line 179. I think this line should contextualize the prevalence of a pandemic and the potential effect on who decides to go to a lab.

Reply: 

* We agree with you. We added sentences discussing the potential selection bias (page 12 line 194 to 197).

Reviewer 3’s Comment (9):

9. Page 17 line 286. It would be informative to provide brief statistics about how many groups were excluded here, and how many participants timed out (none?). A line justifying why full contributions were set as the default (and the inclusion of an automatic program) would be useful too.

Reply: 

* There was no timeout in the experiment and we mentioned this in the beginning of the results. For better understanding, we added a sentence also in this paragraph (page 18 line 306 to 307). In regard to the second point, we explained why the full contribution was set as the default (page 18 line 302 to 304).

Reviewer 3’s Comment (10):

10. Page 19 line 309. It would be useful to know more about such a preliminary experiment.

Reply: 

* We added the information about the preliminary experiment (page 20 line 329 to 331).

Reviewer 3’s Comment (11):

11. Page 21 line 345. Results from the Yahoo crowdsourcing sample currently come a bit out of the blue. I think this sample should be contextualized a bit more before it is introduced here (e.g. by integrating the study as suggested in point 1), and in the Figures. Other than the abstract, no mention of YCrowd is made until this point.

Reply: 

* We appreciate your suggestion. We briefly explained the YCrowd in the Introduction section (page 10 line 154 to 156).

References in this reply letter

Editorial. Tell it like it is. Nature Human Behaviour, 2020; 4 (1), 1.

Bader F, Baumeister B, Berger R, Keuschnigg M. On the transportability of laboratory results. Sociol Methods Res. 2019; 0049124119826151.

Schmelz K, Ziegelmeyer A. Reactions to (the absence of) control and workplace arrangements: experimental evidence from the internet and the laboratory. Exp Econ. 2020; 23(4): 933-60.

Arechar AA, Gächter S, Molleman L. Conducting interactive experiments online. Exp Eco. 2018; 21(1): 99-131.

Fehr E, Gächter S. Altruistic punishment in humans. Nature. 2002; 415(6868), 137-140.

---

## [Editor Report · Decision Letter 2]

6 Apr 2022

Effects of experimental situation on group cooperation and individual performance: comparing laboratory and online experiments

PONE-D-21-25066R2

Dear Dr. Ozono,

We’re pleased to inform you that your manuscript has been judged scientifically suitable for publication and will be formally accepted for publication once it meets all outstanding technical requirements.

Kind regards,

Alfonso Rosa Garcia

Academic Editor

PLOS ONE
---

## [Editor Report · Acceptance letter]

11 Apr 2022

PONE-D-21-25066R2 

Effects of experimental situation on group cooperation and individual performance: comparing laboratory and online experiments 

Dear Dr. Ozono:

I'm pleased to inform you that your manuscript has been deemed suitable for publication in PLOS ONE. Congratulations! Your manuscript is now with our production department. 

Kind regards, 

on behalf of

Dr. Alfonso Rosa Garcia 

Academic Editor

PLOS ONE